# Ropinirole Cotreatment Prevents Perivascular Glial Recruitment in a Rat Model of L-DOPA-Induced Dyskinesia

**DOI:** 10.3390/cells12141859

**Published:** 2023-07-14

**Authors:** Osama F. Elabi, Elena Espa, Katrine Skovgård, Silvia Fanni, Maria Angela Cenci

**Affiliations:** Basal Ganglia Pathophysiology Unit, Department of Experimental Medical Science, Lund University, 221 84 Lund, Sweden; osama.elabi@med.lu.se (O.F.E.); katrine.skovgard@med.lu.se (K.S.); silvia.fanni@med.lu.se (S.F.)

**Keywords:** levodopa-induced dyskinesia, dopamine agonist, neuroinflammation

## Abstract

Dopamine replacement therapy for Parkinson’s disease is achieved using L-DOPA or dopamine D2/3 agonists, such as ropinirole. Here, we compare the effects of L-DOPA and ropinirole, alone or in combination, on patterns of glial and microvascular reactivity in the striatum. Rats with unilateral 6-hydroxydopamine lesions were treated with therapeutic-like doses of L-DOPA (6 mg/kg), an equipotent L-DOPA-ropinirole combination (L-DOPA 3 mg/kg plus ropinirole 0.5 mg/kg), or ropinirole alone. Immunohistochemistry was used to examine the reactivity of microglia (ionized calcium-binding adapter molecule 1, IBA-1) and astroglia (glial fibrillary acidic protein, GFAP), as well as blood vessel density (rat endothelial cell antigen 1, RECA-1) and albumin extravasation. L-DOPA monotreatment and L-DOPA–ropinirole cotreatment induced moderate-severe dyskinesia, whereas ropinirole alone had negligible dyskinetic effects. Despite similar dyskinesia severity, striking differences in perivascular microglia and astroglial reactivity were found between animals treated with L-DOPA vs. L-DOPA–ropinirole. The former exhibited a marked upregulation of perivascular IBA-1 cells (in part CD68-positive) and IBA-1–RECA-1 contact points, along with an increased microvessel density and strong perivascular GFAP expression. None of these markers were significantly upregulated in animals treated with L-DOPA–ropinirole or ropinirole alone. In summary, although ropinirole cotreatment does not prevent L-DOPA-induced dyskinesia, it protects from maladaptive gliovascular changes otherwise associated with this disorder, with potential long-term benefits to striatal tissue homeostasis.

## 1. Introduction

Parkinson’s disease (PD) is characterised by a set of motor symptoms that depend on putaminal dopamine (DA) depletion and respond to pharmacological DA replacement. The most effective replacement therapy is achieved using L-DOPA, a drug that is promptly converted to DA by dopaminergic and serotonergic neurons in the brain [1]. However, the use of L-DOPA is associated with a high risk of motor fluctuations and dyskinesias, whose prevalence increases with treatment duration, reaching approx. 40% by 4–6 years and 90% by 9–15 years (reviewed in [2]). Another approach to pharmacological DA replacement is provided by the so-called DA agonists, a class of drugs that directly stimulate DA receptors without any need for metabolic conversion. The DA agonists currently used for oral administration (such as ropinirole and pramipexole) have a predominant or exclusive activity on dopamine receptors D2/3 and moreover differ from L-DOPA with regard to pharmacokinetics [3]. Pramipexole and ropinirole have similar efficacy and tolerability when used as an adjuvant therapy in advanced stages of PD [4]. Both drugs are clinically less effective than L-DOPA and can be used as monotherapy only in early-mild stages of PD. Although initiating treatment with one of these DA agonists delays the onset of motor complications, dyskinesias develop with the expected incidence when L-DOPA is added to achieve better symptomatic control [5].

At present, the choice of dopaminergic medications at different stages of PD is dictated by the balance between clinical response and side effect risk in individual patients. However, given the profound pharmacological differences that exist between L-DOPA and the DA agonists, choosing one drug or the other (or a combination thereof) may have different functional consequences in the long term. Accordingly, recent studies in PD patients report significant differences in biomarkers of vascular or immune reactivity between subjects treated with L-DOPA monotherapy or L-DOPA–DA agonist combinations [6,7].

Using 6-hydroxydopamine (6-OHDA)-lesioned rats as a parkinsonian model, we have recently defined regimens of L-DOPA monotreatment and L-DOPA–ropinirole cotreatment that have equipotent motor effects [8]. In this experimental paradigm, we have detected different patterns of striatal ∆FosB expression and microvascular plasticity upon L-DOPA–ropinirole cotreatment compared to L-DOPA monotreatment, despite similar levels of dyskinesia.

Here, by applying the same treatment regimens to a new set of animals, we have investigated markers of microglial and glial reactivity in the striatum, with particular attention to changes occurring in the perivascular environment.

## 2. Materials and Methods

### 2.1. Animals

Adult female Sprague Dawley rats (Janvier, France) with a body weight of 250 g on arrival were used in the study. The animals were housed in Innovive cages (San Diego, CA, USA) on a 12 h light/dark cycle with free access to food and water. All procedures were approved by the Malmö-Lund Ethical Committee on Animal Research. 

### 2.2. Study Design

All rats were subjected to 6-OHDA to induce lesions and subsequently allocated to four treatment groups as follows: (1) L-DOPA 6 mg/kg (LD6, *n* = 11); (2) ropinirole 0.5 mg/kg + L-DOPA 3 mg/kg (R0.5 + LD3, *n* = 11); (3) ropinirole 0.5 mg/kg (R0.5, *n* = 9); and (4) saline (*n* = 7). Drug doses were chosen based on [8], to which we refer for a detailed rationale. The treatments were given as daily subcutaneous (s.c.) injections for 28 days. Dyskinesia rating sessions were performed on treatment days 5, 8, 12, and 22. On the last day of treatment, all the animals were sacrificed one hour after the last injection (“on-treatment”).

### 2.3. Dopamine-Denervating Lesions 

Unilateral DA-denervating lesions were obtained by injecting 6-OHDA into the right medial forebrain bundle (MFB) according to a well-established method [9]. Briefly, rats were anesthetized with an intraperitoneal (i.p.) injection of fentanyl (Apoteksbolaget, Uppsala, Sweden, 50 µg/mL) and Dormitor (Apoteksbolaget, Uppsala, Sweden, 1 mg/mL) at a ratio of 20:1. The neurotoxin 6-OHDA hydrochloride (Sigma Aldrich AB, Stockholm, Sweden) was dissolved in 0.2% ascorbate–saline (final concentration, 3.5 µg/µL, free base). The rats were fixed on a stereotaxic frame (David Kopf Instruments, California, USA) and the 6-OHDA solution was injected at a rate of 1 µL/min using the following coordinates (in mm relative to bregma and dural surface): 1st site (2.5 µL): AP −4.0, ML −1.2, DV −7.8, tooth bar in flat skull position; 2nd site (2.0 µL): AP −4.0, ML −0.8, DV −8.0, tooth bar at +3.4. After surgery, rats were given an anaesthesia antidote (Antisedan, Apoketsbolaget, Uppsala, Sweden, 0,1 mg/kg s.c.), and analgesic treatment (Temgesic, Apoketsbolaget, Uppsala, Sweden, 0.01 mg/kg s.c.). In the two weeks following surgery, animals were observed daily, and food supplements were given where appropriate to help them rapidly recover their initial body weight. 

### 2.4. Evaluation of 6-OHDA Lesion Efficacy

Three weeks after surgery, the animals were evaluated in a test of forelimb hypokinesia which measures limb use during vertical exploration of a cylindrical enclosure (cylinder test), according to a previously described protocol [8,10,11]. Only animals with ≤25% contralateral forelimb use were included in this study, thus only selecting rats with severe unilateral DA denervation [11,12]. Moreover, a final verification of lesion extent was carried out by tyrosine hydroxylase (TH) immunostaining of striatal sections. The virtual absence of striatal TH-positive innervation on the side ipsilateral to the lesion was verified by inspecting the striatal sections on a light table (See Appendix A). In cases where residual DAergic fibres were apparent, optical density (O.D.) analysis was applied as in [13]. Only animals with ≥85% overall loss of striatal TH expression on the side ipsilateral to the lesion (relative to the intact side) were included in this study. Animals allocated to the different treatments had similar levels of striatal DA denervation (Appendix A).

### 2.5. Drug Treatments

L-DOPA (L-3,4-dihydroxyphenylalanine methyl ester hydrochloride, Sigma Aldrich AB, Sweden) was always co-administered s.c. with a fixed dose of the peripheral DOPA-decarboxylase inhibitor benserazide (benserazide hydrochloride, 12 mg/kg; Sigma Aldrich AB, Stockholm, Sweden). Ropinirole (Hello Bio, Bristol, UK) was dissolved separately in saline solution and injected s.c., with or without L-DOPA (see treatment groups at 1.2). The selected doses of L-DOPA (6 mg/kg) and ropinirole (0.5 mg/kg) are commonly used in parkinsonian rat models and have been previously shown to produce an equivalent improvement in forelimb hypokinesia in MFB-lesioned rats [8]. The combination of L-DOPA 3 mg/kg and ropinirole 0.5 mg/kg has been previously shown to have anti-akinetic and dyskinetic effects similar to L-DOPA 6 mg/kg [8]. 

### 2.6. Ratings of Abnormal Involuntary Movements 

Ratings of abnormal involuntary movements (AIMs) were carried out using previously described methods [9]. Briefly, rats were individually observed for 1 min every 20 min during 3 h following drug administration. Axial, limb, and orolingual AIMs (see examples in Appendix A) were given separate severity scores using both a time-based scale and an amplitude-based scale, each graded 0 to 4 [14]. On the time-based scale, severity is defined based on the proportion of observation time during which dyskinetic features are present. On the amplitude scale, severity is defined based on the divergence of a dyskinetic body part from its natural resting position. The total AIM score per monitoring period was calculated by multiplying time- and amplitude-based scores for each AIM subtype (axial, limb, and orolingual) in each monitoring period. The AIM score per session was defined by the sum of all these products.

### 2.7. Tissue Preparation and Immunohistochemistry

One hour after the last drug (or saline) injection, the rats were injected with a lethal dose of sodium pentobarbital (240 mg/kg i.p., Apoteksbolaget AB, Uppsala, Sweden) and subjected to transcardial perfusion with a room-temperature physiological saline solution followed by ice-cold 4% paraformaldehyde (VWR) in 0.1 M phosphate buffer (pH 7.4). The brains were then removed, post-fixed in paraformaldehyde solution for 2 h, and cryoprotected in 25% sucrose in 0.1 M phosphate buffer at 4 °C. Subsequently, the brains were cut on a freezing microtome into coronal sections of 40 µm thickness. The sections were then stored at −20 °C in a non-freezing solution (0.5 M sodium phosphate buffer, 30% glycerol, and 30% ethylene glycol). 

Bright-field immunohistochemistry was performed according to well-established methods [8,15] using primary antibodies for tyrosine hydroxylase (TH, rabbit anti-TH antiserum, Peel Freez P40101, 1:1000), and Glial Fibrillary Acidic Protein (GFAP, rat anti-GFAP, Invitrogen, Waltham, MA, USA, 13-0300, 1:1000). Immunocomplexes were revealed using biotinylated secondary antibodies from Vector Laboratories (goat anti-rabbit BA 1000, 1:200, rabbit anti-rat BA 4001, 1:250), followed by an avidin–biotin–peroxidase solution (ABC Elite Kit, Newark, Vector Laboratories, CA, USA). The final chromogenic reaction was developed using 3,3′-diaminobenzidine (DAB) and 0.04% H_2_O_2_ in a buffered solution.

Double or triple immunofluorescence was used to enable simultaneous visualisation of microglia, astrocyte, blood vessels, and albumin. First, free-floating brain sections were blocked in 5% serum diluted in 0.25% Triton-X100-phosphate-buffered saline (PBS-TX) for one hour at room temperature (RT). This was followed by overnight incubation with primary antibodies diluted in 3% serum-containing PBS (2-night incubation at 4 °C was used when staining for RECA-1). The following primary antibodies were used: rat endothelial cell antigen 1 (RECA-1, Bio-Rad, Copenhagen, Denmark cat# MCA970R, 1:100); ionized calcium-binding adapter molecule 1 (IBA-1, Wako, Osaka, Japan, Cat# 019-19741, 1:500); albumin (Bio-Rad, cat# 0220-2424, 1:5000); GFAB (MilliporeSigma, Solna, Sweden, cat# MAB360, 1:400); CD31 (R&D Systems, Minnesota, USA, cat# AF3628, 1:200); and CD68 (Bio-Rad, cat# MCA1957, 1:200). The sections were then incubated with appropriate fluorophore-tagged secondary antibodies for one hour at RT to stain albumin and IBA-1 (Cy3, 1:500 dilution, and Alexa Flour-647, 1:500 dilution, Jackson Immuno-Research, Cambridgeshire, UK, respectively). For RECA-1, a biotinylated secondary antibody (1:200 dilution, Vector-Laboratories, Newark, CA, USA) followed by Alexa Flour 488-conjugated streptavidin (1:500 dilution, Thermofisher, Lund, Sweden) was used to amplify the signal. Finally, the sections were counterstained with DAPI (diamidino-2-phenylindole) (1:1000 in PBS for 10 min), mounted on gelatinized slides, and coverslipped with PVA/DABCO mounting medium.

### 2.8. Confocal Microscopy 

A Leica DMi8 confocal microscope was used to acquire confocal images from two striatal sections per animal encompassing the main body of the striatum (bregma + 0.36 and −0.1, respectively, according to [16]. Sections through the lateral (motor) striatum were visualised under a 20X objective, and sample areas were imaged on the side ipsilateral to the lesion (3 sample areas/section) and the contralateral one (1 sample area/section) using the same acquisition settings (image size: 775 μm × 775 μm; z-stack size = 10 μm; step size = 1 μm) unless specified somewhere else. The same sample areas were used to analyse microglia, vessel density, and albumin leakage.

### 2.9. Quantitative Image Analyses 

Quantitative analyses were carried out in a blinded fashion using computerised methods. 

*Microglia cell analyses:* Total IBA-1^+^ microglia were counted in the maximum projected images acquired under a 20X objective. In the same field of view, perivascular IBA-1^+^ microglia were counted within a 7 µm distance from blood vessel borders (enabling direct physical interaction between the two components). The counting was performed using ImageJ software (version 1.53t, NIH, Bethesda, MA, USA). To count all microglia cells within the sample area, the colocalization tool of ImageJ was used to highlight the colocalization of microglial cell body and nucleus. Then, the highlighted colocalized points were counted using the Analyse Particles tool in ImageJ. The counting process was then verified to exclude microglial branches’ colocalization with the nucleus of other cells, and manual corrections were applied where necessary. The distance from the blood vessel border was determined using the Distance Map tool of ImageJ. Perivascular microglia counts were expressed as a percentage of the total number of microglia cells in the sample area. 

*Microglia interaction with the vessels:* The direct physical interaction of perivascular microglia with adjoining vessels was analysed using the Colocalization plugin tool of ImageJ (version 1.53t, NIH, Bethesda, MA, USA). After determining the image threshold for IBA-1 and RECA-1, the interaction points between IBA-1^+^ cells and RECA-1^+^ vessels were highlighted in each confocal plane of the z-stack image. Then, a maximum projection of the interaction points was created and the total density of these contact points was measured using the area fraction measurement tool of ImageJ (version 1.53t, NIH, USA). The data were expressed as the density of interaction points normalized to the blood vessel density in the sample area. To obtain a three-dimensional visualization of microglial–vascular topology, Imaris software (version 8.0, Bitplane, Zurich, Switzerland) was used to render z-stack images of RECA-1 and IBA-1 using the surface tool wizard. 

*Vessel density measurement:* RECA-1^+^ vessel density was quantified on the maximum projected images. By applying the image threshold, the density of the vessels per image was measured using the area fraction measurement tool of ImageJ (version 1.53t, NIH, Bethesda, MA, USA). 

*Albumin leakage:* Parenchymal albumin expression was quantified in the maximum projected images acquired under a 20X objective. After applying a constant threshold value, the number of albumin-positive pixels was calculated using the area fraction measurement tool of ImageJ (version 1.53t, NIH, USA). The data were expressed as albumin density in the lesioned side relative to the values measured on the intact side of the same section. 

*GFAP:* The expression of GFAP was quantified through bright-field microscopy using a Nikon 80i microscope connected with a digital camera (Olympus DP72), supporting an x-y motorized stage guided by NewCAST software (version 4.4.4.0, Visiopharm, DK) (see [15,17] for additional details). An image segmentation software (VIS, version 4.4.4.0, Visiopharm, DK) was trained to discriminate GFAP^+^ pixels from the background using a Bayesian algorithm-based pixel classification method. Four rostrocaudal levels through the lateral striatum (Bregma: +0.36, +0.2, −0.1, and −0.36, as in [16]) were analysed for each animal. Sample areas (acquired under a 20X objective) were randomly chosen by the software so as to cover 60% of the lateral striatum. The data were expressed as the average number of GFAP^+^ pixels per sample area.

*Perivascular and parenchymal CD68-positive microglia*: To examine the activation state of microglia, a sample of animals (*n* = 7 per treatment group) was processed for triple-immunostaining for the vascular marker CD31, the microglial marker IBA-1, and cluster of differentiation 68 (CD68), which is expressed by phagocytically activated microglia and macrophages [18]. Confocal images were acquired under a 40X objective. Z-stack images (acquisition settings: images size 378 μm × 378 μm; z-stack size = 5 μm; step size = 0.45 μm) were converted to a maximum projection image before the analysis. First, perivascular IBA-1^+^ microglia (located within 7 µm from the CD31^+^ blood vessel border) were distinguished from the parenchymal IBA-1^+^ microglia (located at more than 7 µm from blood vessel border) using the Distance Map tool of ImageJ (version 1.53t, NIH, Bethesda, MA, USA). Then, by using the colocalization tool of ImageJ, the density of CD68 cellular staining colocalized with IBA-1 was quantified and expressed as a percentage of IBA-1 cell density in the perivascular or parenchymal space, respectively. Imaris software (version 8.0, Bitplane, Zurich, Switzerland) was used to render z-stack images of CD31, CD68, and IBA-1 staining using the surface tool wizard. A transparent mode was used to render IBA-1, enabling visualizing the colocalized CD68. 

### 2.10. Statistical Analysis

Statistical analysis was performed using Prism 9 (GraphPad Software, version 10.0.0, San Diego, CA, USA). When comparing treatment effects over time (AIM scores), mixed-effect ANOVA was used as it allows for the evaluation of the interaction between time and treatment. Post hoc analyses were carried out using Bonferroni’s multiple comparison test. Treatment effects on AIM subscores were examined using non-parametric Kruskal–Wallis tests followed by Dunn’s post hoc tests. Histopathological data were analysed using the Kruskal–Wallis test and the Mann–Whitney test was used for post hoc pairwise comparisons where appropriate. Two-way repeated measures ANOVA followed by a Bonferroni’s post hoc test was used to compare CD68-positive microglia counts between perivascular vs. parenchymal locations and treatment groups. Linear regression was used to analyse relationships between the immunohistochemical data and AIM scores (using values from the last test session in the chronic treatment). In all analyses, statistical significance was set at *p* < 0.05. 

## 3. Results

### 3.1. L-DOPA Monotreatment and L-DOPA–Ropinirole Cotreatment Had Equipotent Dyskinetic Effects

Chronic administration of LD6 and R0.5 + LD3 caused a development of moderate-severe dyskinesia, with a comparable overall severity and time course (Figure 1A,B) and a similar body distribution (Figure 1C) in the two treatment groups. Among the R0.5 + LD3-treated animals, there was a trend towards a larger representation of axial relative to orolingual AIM scores (Figure 1C), but the difference from LD6-treated animals did not reach statistical significance. In keeping with previous reports [8,19,20], de novo treatment with ropinirole alone had a very low dyskinesiogenic action, and the ropinirole monotreatment group did not differ from saline-treated animals (see red vs. empty symbols in Figure 1A,B). 

### 3.2. Differential Effects of L-DOPA and Ropinirole on Microglial Cells and Their Vascular Association

Vascular plasticity and pro-inflammatory glial responses have been linked to LID by several previous studies [6,7,15]. We therefore evaluated the effects of LD6, R0.5 + LD3, and R0.5 on the number of microglial cells and their relationship with blood vessels in the lateral (motor) part of the striatum. The number of IBA-1^+^ cells was significantly increased in LD6-treated animals compared to all the other groups, whereas animals treated with R0.5 +LD3 or R0.5 alone did not differ significantly from saline-treated controls (Figure 2A’–A’’’,D). 

Next, we counted the number of microglial cells located in close proximity to the blood vessels and expressed these data as a percentage of perivascular microglia relative to the total microglial counts in the same sample area. In the LD6-treated group, up to approx. 55% of microglial cells were localised in close proximity to the blood vessels (*p* < 0.05 for LD6 vs. all other groups; Figure 2B–B’’’,C–C’’’,E). Interestingly, rats treated with R0.5, alone or combined with LD3, showed a low percentage of perivascular microglia (median value 10–12%), even lower than that found in saline-treated controls (*p* < 0.05 for R0.5 + LD3 and R0.5 vs. saline; Figure 2B’’,B’’’,C’’,C’’’,E). 

To determine the extent of physical interactions between microglial processes and vascular structures, we searched for points of colocalization between IBA-1 and RECA-1 immunofluorescence. The rats treated with LD6 exhibited a larger extent of IBA-1–RECA-1 colocalization than did all the other groups (*p* < 0.05; Figure 2F). Interestingly, treatment with R0.5, alone or in combination with LD3, reduced the extent of microglial/vascular interactions even below the levels measured in saline-treated controls (*p* < 0.05 for R0.5 + LD3 and R0.5 vs. saline; Figure 2B”,B’’’,C”,C’’’,F). Linear regression analysis showed that perivascular microglial cell numbers and IBA-1–RECA-1 interaction points were positively correlated with the AIM scores considering all treatment groups (R = 0.38 and 0.39, respectively, *p* < 0.05; Appendix A).

We next examined the microglial expression of CD68, a protein primarily localised to lysosomes and endosomes in phagocytically active cells of the monocyte lineage [18]. In LD6-treated animals, a noticeable number of microglial cells in the perivascular region displayed intracellular positivity for CD68 (Figure 3B,E; *p* < 0.05 for LD6 vs. all other groups). In contrast, animals treated with R0.5 + LD3 or R0.5 exhibited low perivascular expression of CD68-positive IBA-1 cells, not differing from saline-treated controls (Figure 3C–E). In both perivascular and parenchymal locations, the proportion of microglial cells positive for CD68 correlated positively with the AIM scores when considering all treatment groups (R = 0.49, *p* < 0.05; Appendix A).

### 3.3. Treatment Effects on Vascular Density and BBB Permeability 

We have shown previously that LD6, R0.5 + LD3, and R0.5 treatments have different effects on markers of vascular integrity and angiogenesis [8]. We therefore examined the effects of these treatments on vessel density and vascular leakage in the same sample areas used to study the microglia. The quantification of RECA-1 immunofluorescence demonstrated that treatment with LD6 had resulted in increased vascularization of the striatal tissue (*p* < 0.05 vs. saline; Figure 2B’,C’,G). In contrast, treatment with R0.5 + LD3 or R0.5 did not significantly alter striatal vessel density relative to saline control values (Figure 2B’’,B’’’,G). 

To evaluate BBB permeability, we assessed the parenchymal expression of albumin, a plasma protein. The area fraction positive for albumin (albumin density) was significantly increased in animals treated with LD6 compared to all other groups (Figure 4B vs. Figure 4A,C,D, and data in Figure 4E). In contrast, albumin density remained within control values in animals treated with R0.5 + LD3 or R0.5 (Figure 4A,C,D, and data in Figure 4E). A 3D-rendering of albumin, RECA-1, and IBA-1 co-staining in animals from the LD6 group revealed a close association of microglial processes with leaky vessels (Figure 4F,G). Both RECA-1 density and albumin density positively correlated with the AIM scores when considering all treatment groups together (R = 0.43 and R = 0.37, respectively, *p* < 0.05; Appendix A).

### 3.4. Elevated Astroglial Reactivity after L-DOPA Monotreatment

Previous studies have shown that dyskinesiogenic treatment with L-DOPA results in increased striatal expression of GFAP, an astroglial intermediate filament protein that becomes upregulated in reactive astrocytes [21,22,23]. We therefore explored possible group differences in the expression of this marker. As shown in Figure 5, LD6 monotreatment induced a marked GFAP upregulation (Figure 5B compared to Figure 5A,C,D), differing significantly from all other treatment conditions (*p* < 0.05 for LD6 vs. all other groups, Figure 5E). In animals treated with LD6, the strongest expression of GFAP was seen in close proximity to blood vessel profiles (Figure 5B), and several vessels were found to be physically enwrapped by thick GFAP-positive processes (Figure 5F–F’’). 

## 4. Discussion

L-DOPA-induced dyskinesia is causally linked with maladaptive plastic changes to both neurons and non-neuronal cells in the basal ganglia network [1]. Abnormal microvascular plasticity is found in striatal, pallidal, and nigral regions in parkinsonian animals that develop dyskinesia upon chronic L-DOPA treatment [15,17,24,25,26,27,28]. This microvascular plasticity involves focal areas of angiogenesis and BBB hyperpermeability and appears to be related to an abnormal vasomotor response to L-DOPA in the affected areas in vivo [15,27]. The clinical relevance of these findings is supported by functional imaging studies in PD patients affected by LID [29,30], as well as post-mortem immunohistochemical observations [15]. A causal link between L-DOPA-induced microvascular plasticity and dyskinesia has been demonstrated using pharmacological inhibitors of angiogenic cytokine signalling. Animals receiving these anti-angiogenic treatments develop less severe dyskinesia over a chronic course of L-DOPA treatment [15,31]. A similar preventive effect has been obtained using immunomodulatory drugs with anti-angiogenic and anti-inflammatory properties [24].

To the best of our knowledge, this is the first study showing that (a) L-DOPA-induced angiogenesis and BBB leakage go hand-in-hand with perivascular recruitment of reactive microglia and astroglia and, (b) this perivascular glial reactivity is prevented by cotreatment with ropinirole, which also inhibits dyskinesia-associated angiogenesis and BBB hyperpermeability, though not the expression of AIMs. Indeed, we found that cotreatment with ropinirole and a sub-dyskinetic dose of L-DOPA (3 mg/kg) [8,32] induced the same type of moderate-severe dyskinesia as did a full-dose (6 mg/kg) L-DOPA regimen. These results are congruent with the clinical observation that ropinirole and other D2/3 agonists do not reduce the incidence nor the severity of dyskinesia when given together with L-DOPA [33,34] (which possibly reflects the cooperativity between D1 and D2/3 receptors in eliciting AIMs [14,35]). Furthermore, these results are in keeping with our recent study reporting similar AIM scores upon L-DOPA-only or L-DOPA–ropinirole administration at the same dosages used here. In further agreement with our previous study, we found here that ropinirole monotreatment induced only mild and occasional signs of dyskinesia, not differing significantly from vehicle (saline) as to the total AIM score per test session. An important methodological difference should, however, be noted: animals in the previous study had received the L-DOPA–ropinirole combination after chronic exposure to either L-DOPA or ropinirole alone [8]. In contrast, all animals in the present study received their treatments de novo. Furthermore, in the previous study, brains were collected for histological analyses 24 h after the last drug/vehicle injection, whereas rats in the present study were sacrificed while “on treatment” (60 min post-injection). These technical details are highlighted here because they indicate that the observed differences between L-DOPA and L-DOPA–ropinirole treatment are not contingent on using a particular experimental design. 

In our previous study, dyskinetic rats under L-DOPA monotreatment were found to express high levels of microvascular nestin immunoreactivity (an angiogenesis marker) as well as albumin leakage in the striatum, whereas rats developing dyskinesia upon L-DOPA–ropinirole cotreatment exhibited low levels of nestin and albumin immunoreactivity, not differing significantly from saline-treated controls in this regard [8]. Also, in the present study, treatment-induced microvascular changes (i.e., increased vessel density and albumin extravasation) were prominent upon L-DOPA monotreatment but mild or absent upon treatment with ropinirole–L-DOPA or ropinirole alone. Using intravenously injected tracers with varying molecular weights would be needed to precisely determine the type and degree of BBB dysfunction [36]. However, the large difference in parenchymal albumin immunostaining between the treatments provides rather clearcut evidence that ropinirole is able to prevent the increase in BBB permeability induced by L-DOPA. 

A new and original contribution of the present study is the demonstration of a strong perivascular glial reactivity upon treatment with L-DOPA, which is prevented by ropinirole cotreatment. Indeed, rats treated with L-DOPA were found to exhibit a high number of microglial cells in close proximity to the microvessels, prominent physical interactions between the microglial process and vessel profiles, and strong astroglial reactivity adjoining blood vessels in the striatum. Furthermore, following L-DOPA treatment, a noticeable number of perivascular microglia cells exhibited an active phagocytic phenotype, as indicated by their intracellular expression of the phagocytic marker CD68. None of the above features occurred in animals treated with ropinirole, either alone or combined with L-DOPA. Interestingly, ropinirole monotreatment tended to reduce all the above parameters even below the levels found in drug-naive 6-OHDA-lesioned animals. 

The mechanisms underlying the L-DOPA-dependent attraction of microglial cells to blood vessels remain to be defined. However, it seems logical to propose that the prime triggering factor comes from the vessels themselves. Importantly, endothelial cells and pericytes in the brain express high levels of DOPA- and DA-metabolising enzymes, providing the first site where exogenous L-DOPA can be converted to DA [1]. Although peripheral DOPA decarboxylase inhibitors (benserazide, carbidopa) are included in the common PD medications, they cannot impede an endothelial conversion of L-DOPA once the drug is transported to the abluminal side of the vessels. At least in vitro, endothelial cells can express DA receptors and release cytokines [37,38,39]. Furthermore, studies in cultured human endothelial cells have reported opposite modulatory effects upon stimulation of D1- vs. D2-class receptors, and the latter have been shown to inhibit cAMP production and angiogenic cytokine signalling [37,38]. Such an inhibitory effect could account for the blockade of L-DOPA-induced angiogenic activity and BBB leakage by ropinirole, given its strong stimulatory effect on D2 and D3 receptors [40]. 

Pericytes are embedded in the basement membrane of small vessels and capillaries, where they play a key role in BBB maintenance, regulation of vessel diameter, as well as structural vessel remodelling under a number of pathophysiological conditions [41]. Recent studies in rodent models of brain disease have detected a positive correlation between the extent of microglial–blood vessel interactions and microvascular pericyte coverage [42,43]. This may depend on the pericytic capacity for cytokine and chemokine secretion, possibly affecting glial reactivity and migration [44]. However, it is currently unknown whether striatal pericytes express DA receptors. 

Downstream to endothelial cells and pericytes, astrocytes are the third vessel-associated cell sensing the entry of dopaminergic drugs into the brain parenchyma. Astrocytes express both D1- and D2-class receptors [45], and they can do so in the striatum after acquiring a reactive state [46]. We show here that L-DOPA monotreatment was associated with a reactive state of striatal astrocytes, as indicated by the upregulation of the intermediate filament protein GFAP, a marker of astrocytic hypertrophy [47]. Evidence linking astrocytic reactivity to DA receptor stimulation on astrocytes has not yet been produced. However, studies in other disease models have shown that astrocytes strongly react to extravasal albumin, which promptly induces maladaptive changes in astroglial function and structure [48,49]. Along with these reports, the fact that high GFAP expression was found in astrocytic processes enwrapping blood vessels suggests that the observed astrocytic reactivity was secondary to the angiogenesis and BBB hyperpermeability induced by L-DOPA monotreatment. Accordingly, animals treated/co-treated with ropinirole showed a concomitant lack of angiogenesis, albumin extravasation, and astroglia reactivity. We therefore propose that ropinirole exerted its primary protective action by activating D2/3 receptors on vessel-associated cells. In line with this hypothesis, the D2/3 agonist pramipexole has been found to reduce BBB damage and increase endothelial tight junction proteins in a traumatic brain injury model [50].

It remains to be established whether the perivascular glial reactivity induced by L-DOPA represents a useful response to limit microvessel damage or rather a maladaptive response contributing to tissue dyshomeostasis and inflammation. Previous studies reporting increased astroglial and microglial reactivity in animal models of LID have linked the observed increase in glial markers to a pro-inflammatory state [21,23,51]. In these studies, the treatment-induced glial reactivity was found to concur with an increased expression of proinflammatory mediators, such as nitric oxide synthase [21], tumour necrosis factor-alpha [51], or interleukin 1-beta [28] in the affected striatal and pallidal regions. Furthermore, a number of anti-inflammatory strategies have been reported to dampen the development of dyskinesia in parkinsonian rodent models treated with L-DOPA (partly reviewed in [52]). Finally, studies in other disease models have shown that the perivascular microglial clustering induced by extravasal blood proteins contributes to neuroinflammation and neuritic damage [53]. It is therefore likely that the perivascular glial recruitment induced by L-DOPA has maladaptive effects, augmenting proinflammatory reactions in the disease-affected striatum and altering the physiological homeostatic functions of glial cells in this region.

## 5. Concluding Remarks

Clinical trials in early-stage PD have suggested that de novo treatment with ropinirole is associated with a slower progression of the disease [54]. Our results reveal a novel beneficial effect of this D2/3 agonist, consisting of the inhibition of L-DOPA-induced angiogenesis, BBB leakage, and the associated perivascular glial recruitment. By inhibiting these maladaptive changes, adjuvant treatment with ropinirole may help protect brain tissue homeostasis over the course of L-DOPA pharmacotherapy even in advanced stages of PD. Studies investigating this possibility in additional animal models as well as human patients are clearly warranted. 

## Figures and Tables

**Figure 1 cells-12-01859-f001:**
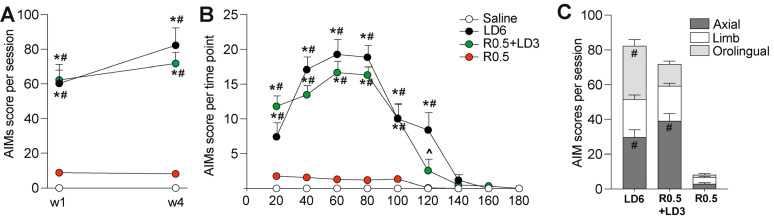
**Dyskinesia profiles in the different treatment groups.** The following treatment groups were studied: L-DOPA 6 mg/kg (LD6, *n* = 11), ropinirole 0.5 mg/kg combined with L-DOPA 3 mg/kg (R0.5 + LD3, *n* = 11), ropinirole alone 0.5 mg/kg (R0.5, *n* = 9), and saline (vehicle control) (*n* = 7). (**A**) Comparison of AIM scores in the first (w1) and last week of treatment (w2). Mixed effect ANOVA: treatment, F(3,33) = 52.88 *p* < 0.0001; time F(1,11) = 2.186 *p* = 0.1673; interaction F(3,13) = 1.157 *p* = 0.3635. Bonferroni’s multiple comparison test: * *p* < 0.005 vs. saline, # *p* < 0.005 vs. R0.5. (**B**) Time course of AIMs during the last test session (treatment day 22). Mixed effect ANOVA: treatment, F(3,30) = 41.30 *p* < 0.0001; time F(8,80) = 51.09 *p* < 0.0001; interaction F(24,195) = 17.12 *p* < 0.0001. Bonferroni’s multiple comparison test: * *p* < 0.005 vs. saline, # *p* < 0.005 vs. R0.5, ^ *p* < 0.05 vs. LD6. (**C**) Representation of axial, limb, and orolingual AIM subscores during the last test session. Kruskal–Wallis tests: KW = 70.75, *p* < 0.0001. Dunn’s multiple comparison test: # *p* < 0.005 vs. R0.5.

**Figure 2 cells-12-01859-f002:**
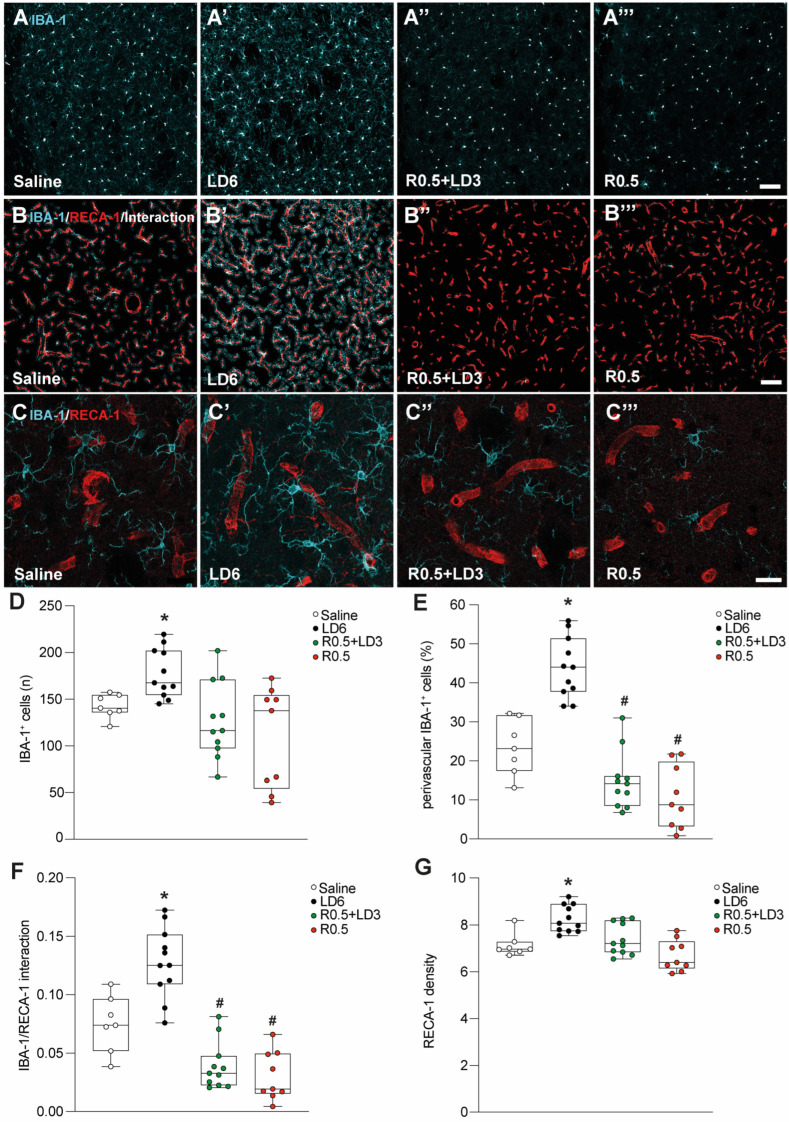
**Microglial cell number and peri-vascular distribution.** Saline (**A**,**B** and **C**, *n* = 7), L-DOPA 6 mg/kg (**A’**,**B’** and **C’**, LD6, *n* = 11), the combination of ropinirole 0.5 mg/kg with L-DOPA 3 mg/kg (R0.5 + LD3, **A’’**,**B’’** and **C’’**, *n* = 11), and ropinirole monotreatment 0.5 mg/kg (R0.5, **A’’’**,**B’’’** and **C’’’**, *n* = 9) were injected chronically and received one hour prior sacrifice. (**A**–**A’”**) Processed confocal images showing the number of IBA-1^+^ microglia (cyan) with highlighted cell body (white). Scale bar = 100 µm. (**B**–**B’’’**) Processed confocal images showing IBA-1^+^ cells (cyan) around RECA-1^+^ vessels (red) within a 7 μm distance from the vessel, and the interaction points between the microglia and the vessels (white). Scale bar = 100 µm. (**C**–**C’’’**) Confocal images acquired at high magnification (40X) illustrate the location of IBA-1^+^ microglia (cyan) in relation to the RECA-1^+^ vessels (red). Scale bar = 20 µm. (**D**) Quantification of IBA-1^+^ cell count in the lesioned side of the striatum; Kruskal–Wallis tests: KW = 12.70, *p* = 0.0053. Mann–Whitney test: * *p* < 0.05 vs. all other groups. (**E**) Histogram showing the percentage of perivascular IBA-1^+^ cell count over the total IBA-1^+^ cell count in the lesioned side of the striatum; Kruskal–Wallis tests: KW = 27.0, *p* < 0.0001. Mann–Whitney test: * *p* < 0.05 vs. all other groups, # *p* < 0.05 vs. saline. (**F**) Data illustrating the density of the interaction points between IBA-1^+^ cells and RECA-1^+^ vessels (fraction of IBA-1^+^ staining colocalized with RECA-1 and normalized to the vessel density); Kruskal–Wallis tests: KW = 27.33, *p* < 0.0001. Mann–Whitney test: * *p* < 0.05 vs. all other groups, # *p* < 0.05 vs. saline (**G**) Histogram showing the density of RECA-1^+^ vessels in the ipsilateral side of the striatum; Kruskal–Wallis tests: KW = 17.41, *p* = 0.0006. Mann–Whitney test: * *p* < 0.05 vs. all other groups.

**Figure 3 cells-12-01859-f003:**
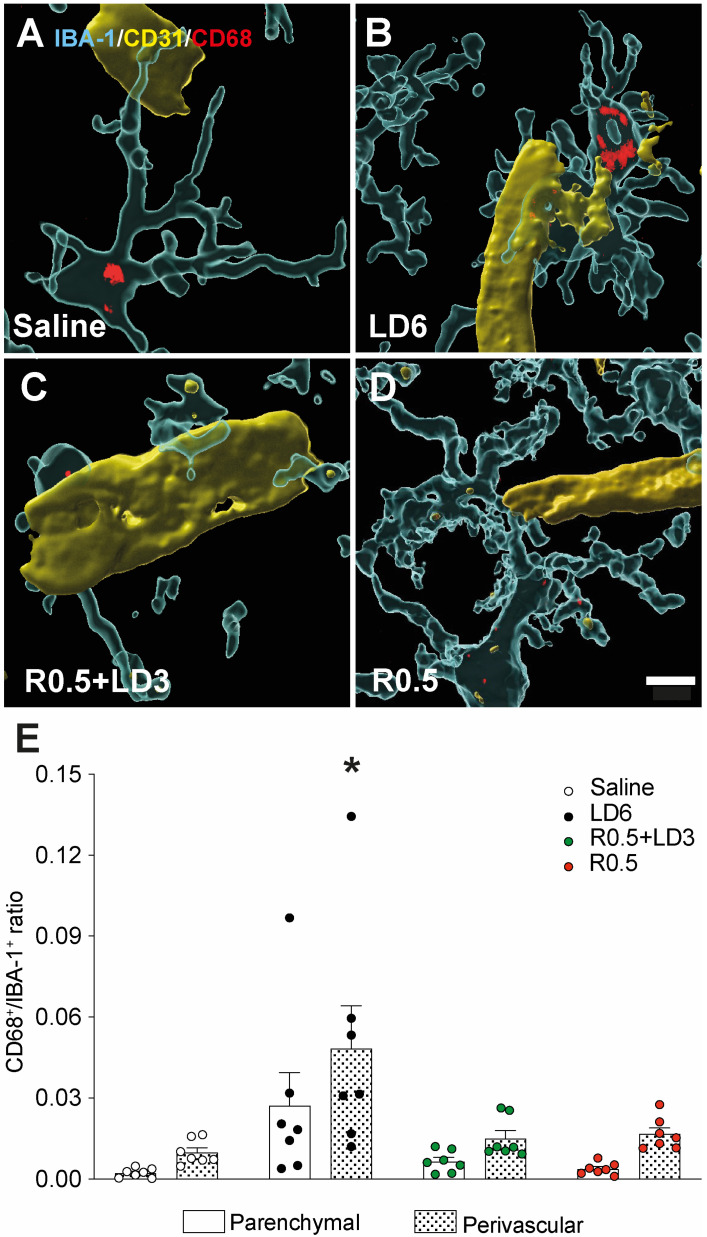
**CD68-positive microglia cells and their peri-vascular distribution in the striatum.** (**A**–**D**) 3D modelling of confocal images rendered by Imaris software show CD68 (red) within microglia cell bodies (IBA-1, transparent cyan), and the proximity of CD68-positive microglia with CD31^+^ vessels (yellow) in animals treated with saline (**A**), LD6 (**B**), R0.5 + LD3 (**C**), and R0.5 (**D**). Scale bar = 5 µm. (**E**) Quantification of CD68-positive microglia, expressed as a ratio of the total IBA-1^+^ cell numbers. Two-way RM ANOVA: treatment, F(3,24) = 4.256 *p* = 0.0152; localization F(1,24) = 63.08 *p* < 0.0001; interaction F(3,24) = 3.799 *p* = 0.0232. Bonferroni’s multiple comparison test: * *p* < 0.005 vs. perivascular counts in all the other groups.

**Figure 4 cells-12-01859-f004:**
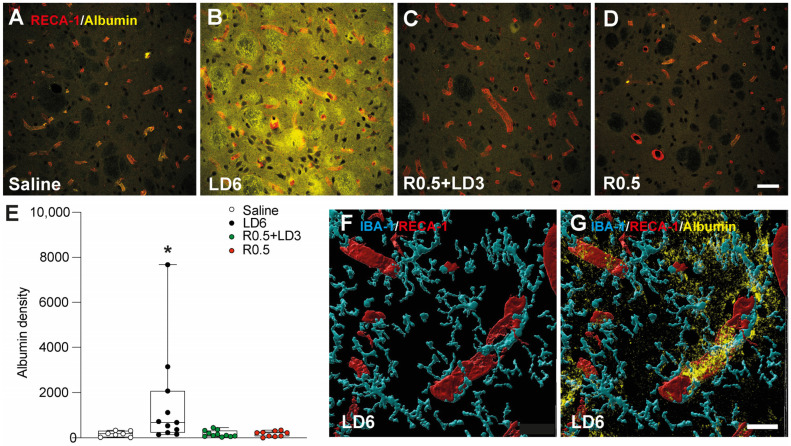
**L-DOPA and ropinirole have quite different effects on striatal BBB integrity.** Saline (**A**, *n* = 7), L-DOPA 6 mg/kg (LD6, **B**
*n* = 11), the combination of ropinirole 0.5 mg/kg with L-DOPA 3 mg/kg (R0.5 + LD3, **C**, *n* = 11), and ropinirole 0.5 mg/kg (R0.5, **D**, *n* = 9) were administered chronically and the last injection was given one hour prior to sacrifice. (**A**–**D**) Confocal images show extravasation of albumin (yellow) from RECA-1^+^ vessels (red) in the DA-denervated striatum. Scale bar = 50 µm. (**E**) Quantification of albumin density (expressed as a percentage of that measured on the intact side of the striatum). Kruskal–Wallis tests: KW = 12.89, *p* = 0.0049. Mann–Whitney test: * *p* < 0.05 vs. all other groups. A 3D model of confocal images acquired at 40X magnification and rendered by Imaris software showing RECA-1^+^ vessel (red) surrounded and enwrapped by IBA-1+ microglia (cyan) (**F**) and the presence of perivascular albumin leakage (yellow) (**G**). Scale bar = 20 µm.

**Figure 5 cells-12-01859-f005:**
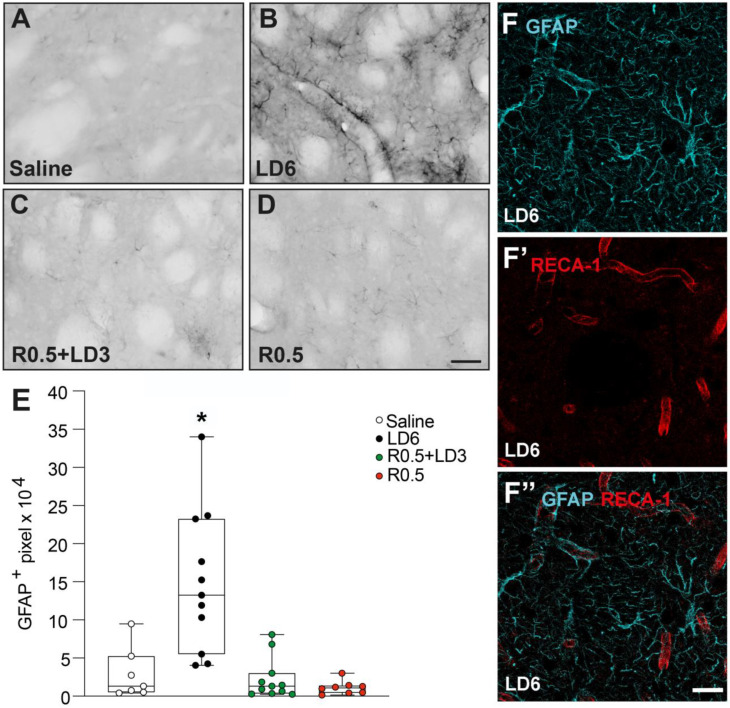
**GFAP expression induced by the different dopaminergic treatments.** GFAP immunohistochemistry was quantified in the lateral part of the striatum (**A**–**D**) in rats receiving chronic treatment with saline (**A**, *n* = 7), L-DOPA 6 mg/kg (LD6) (**B**, *n* = 11), the combination of ropinirole 0.5 mg/kg with L-DOPA 3 mg/kg (R0.5 + LD3) (**C**, *n* = 11), or ropinirole alone 0.5 mg/kg (R0.5) (**D**, *n* = 9). Scale bar = 50 μm. (**E**) Quantification of GFAP-positive pixels. Kruskal–Wallis tests: KW = 20.08, *p* = 0.0002. Mann–Whitney test: * *p* < 0.005 vs. all other groups. (**F**–**F’’**) Photomicrographs showing GFAP^+^ cells (cyan) (**F**), RECA-1^+^ vessels (red) (**F’**), and merged images showing GFAP-positive astrocytes surrounding RECA-1 positive blood vessels (**F’’**). Scale bar = 20 μm.

## Data Availability

Data available on request due to restrictions e.g., privacy or ethics. The data presented in this study are available on request from the corresponding author.

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
