# Peer review of "Ropinirole Cotreatment Prevents Perivascular Glial Recruitment in a Rat Model of L-DOPA-Induced Dyskinesia"

_cells, 2023, doi:10.3390/cells12141859_

Round 1

Reviewer 1 Report

This article reports original and important data on the pathophysiology of L-DOPA-induced dyskinesia (LID) and its prevention by ropinirol, a D2/D3 agonist, in an animal model  of LID,  unilateral 6-OH-dopamine-lesioned rats. The results showed that ropinirol prevents maladpitve plastic changes in both neurons and non-neuronal celles, astrocytes and pericytes, in the basal ganglia regions.  Abnormal microvascular plasticity and blood-brain barrier (BBB) leakage in striatal, pallidal, and nigral regions in this animal models are original findings. 

1) The doses of L-DOPA and ropinirol are important to create the experimental conditions closely similar to L-DOPA-induced dyskinesia condiitons in Parkinsonian patients, and to the preventive effect of ropinirol in Parkinsonian patients. The doses of L-DOPA at 6mg/kg,  ropinirol at 0.5 mg/kg, and L-DOPA-ropinirol combination at 3mg/kg and 0.5 mg/kg, were determined based on the precise examination zin the previous investigation in Refeence 7. However, a brief explanation on the determination of the doses should be described.

2) Possibility in differences in the degrees of responces in humans and rats should be discussed.

3) The reason why ropinirol among D2- D3-agonists is chosen is thought to be based on the clinical efficacy , which should be described.

Author Response

1) The doses of L-DOPA and ropinirol are important to create the experimental conditions closely similar to L-DOPA-induced dyskinesia condiitons in Parkinsonian patients, and to the preventive effect of ropinirol in Parkinsonian patients. The doses of L-DOPA at 6mg/kg,  ropinirol at 0.5 mg/kg, and L-DOPA-ropinirol combination at 3mg/kg and 0.5 mg/kg, were determined based on the precise examination zin the previous investigation in Refeence 7. However, a brief explanation on the determination of the doses should be described.

1- R: Thank you for this valuable comment. We have now added a short paragraph to Material & Methods (point 2.5) to explain the rationale behind the chosen drug doses.

2) Possibility in differences in the degrees of responces in humans and rats should be discussed.

2-R: Thank you for inviting us to comment upon the clinical relevance of our experimental findings. We are doing it in two newly added passages in the Discussion [see: “The clinical relevance of these findings is supported by functional imaging studies in PD patients affected by LID (Jourdain et al. 2016, 2017), as well as post-mortem immunohistochemical observations (Ohlin et al. 2011)”;  “These results are congruent with the clinical experience, as ropinirole and other DA agonists do not mitigate the development nor the expression of dyskinesia when given together with L-DOPA (Stowe et al 2011; Grigoriou et al. 2023)”].

     Regarding the preventive action of ropinirole on maladaptive gliovascular plasticity, no information is currently available on this potential action of ropinirole in the human PD brain. We have therefore added this interesting new sentence to the concluding remarks, pointing to the need for further studies: “Studies investigating this possibility in additional animal models as well as human patients are clearly warranted”.

3) The reason why ropinirol among D2- D3-agonists is chosen is thought to be based on the clinical efficacy , which should be described.

3-R: We chose ropinirole over the closely related drug pramipexole not because it is more efficacious, but because it has a slightly faster elimination half-life (6 hours instead of 8 hours, reviewed in Cenci et al 2011). We have now added this specification to the Introduction “Pramipexole and ropinirole have similar efficacy and tolerability as adjuvant therapy in advanced stages of PD (Zhao et al. 2019). Both drugs are clinically less effective than L-DOPA…”.

Reviewer 2 Report

This is an interesting study showing that L-DOPA 16 monotreatment and L-DOPA-ropinirole cotreatment induced moderate-severe dyskinesia, whereas ropinirole alone had negligible dyskinetic effects. Despite similar dyskinesia severity, striking differences in perivascular microglia and astroglial reactivity were found between animals treated with L-DOPA vs. L-DOPA-ropinirole. The former exhibited a marked upregulation of perivascular IBA-1 cells and IBA-1-RECA-1 contact points, along with an increased microvessel density and strong perivascular GFAP expression. None of these markers were significantly upregulated in animals treated with L-DOPA-ropinirole or ropinirole alone.

An important issue however, is if there are functional changes in BBB permeability correlated with the histochemical changes seen here. While albumin leakage is indirect, BBB leakage from periphery into the CNS is also important. While there are sophisticated techniques using radio traces etc, there are also some very simple approaches using e.g. intravenous injection of dyes and then carrying out routine histology of brain tissue. These are summarized in: Smith Q. A review of blood-brain barrier transport techniques. In: Nag S, editor. The Blood-Brain Barrier: Biology and Research Protocols. Humana; Totowa, NJ: 2003. pp. 193–207. Smith QR, Allan DD. In situ brain perfusion technique. In: Nag S, editor. The Blood-Brain Barrier: Biology and Research Protocols. Humana; Totowa, NJ: 2003. pp. 209–18. Takasato Y, Rapoport SI, Smith QR. An in situ brain perfusion technique to study cerebrovascular transport in the rat. Am. J. Physiol. Heart Circ. Physiol. 1984;247:484–93.

An additional point is the histochemical appearance of the microglia in terms of resting vs activated states. Are there any observations of microglia morphology in terms of cell bodies and processes in the 2 groups?

Finally, were there any controls for the ICC in terms of omission of primary antibodies and blinded observers. Additional data on these points would strengthen this study in terms of the significance of the observations reported here.

Author Response

    An important issue however, is if there are functional changes in BBB permeability correlated with the histochemical changes seen here. While albumin leakage is indirect, BBB leakage from periphery into the CNS is also important. While there are sophisticated techniques using radio traces etc, there are also some very simple approaches using e.g. intravenous injection of dyes and then carrying out routine histology of brain tissue. These are summarized in: Smith Q. A review of blood-brain barrier transport techniques. In: Nag S, editor. The Blood-Brain Barrier: Biology and Research Protocols. Humana; Totowa, NJ: 2003. pp. 193–207. Smith QR, Allan DD. In situ brain perfusion technique. In: Nag S, editor. The Blood-Brain Barrier: Biology and Research Protocols. Humana; Totowa, NJ: 2003. pp. 209–18. Takasato Y, Rapoport SI, Smith QR. An in situ brain perfusion technique to study cerebrovascular transport in the rat. Am. J. Physiol. Heart Circ. Physiol. 1984;247:484–93.

R. We thank the Reviewer for this suggestion, which has prompted us to add the following comment to the Discussion: Using intravenously injected tracers with varying molecular weights would be needed to precisely determine the type and degree of BBB dysfunction (Takasato et al. 1984). However, the large difference in parenchymal albumin immunostaining observed between treatments provides clearcut evidence that ropinirole is able to prevent the increase in BBB permeability induced by L-DOPA”.

An additional point is the histochemical appearance of the microglia in terms of resting vs activated states. Are there any observations of microglia morphology in terms of cell bodies and processes in the 2 groups?

R: We have opted not to rely on standard morphological criteria of microglia state classification because these are currently called into question (see DOI: 10.1016/j.neuron.2022.10.020). Indeed, reactive and surveying microglial cells can have extensive branches, whereas ameboid microglia can display reduced phagocytosis (see DOI: 10.1016/j.neuron.2022.10.020). In response to the Reviewer´s comment we have instead carried out a new round of triple immunostainings including antibodies against CD68 (marker of phagocytically activated cells of the monocyte lineage). By quantifying the proportion of IBA-1 cells that stained positively for CD68, we could reveal a significant difference between animals treated with LD6 versus all the other groups. These new analyses have resulted in the addition of new paragraphs to Material and Methods and Results, and additional phrases or sentences in Abstract and Discussion (all being yellow-highlighted in the manuscript).

Finally, were there any controls for the ICC in terms of omission of primary antibodies and blinded observers. Additional data on these points would strengthen this study in terms of the significance of the observations reported here.

R: All our immunohistochemical methods have been established and optimised through extensive pilot experiments (including serial dilutions of primary and secondary antibodies) before applying them to the entire experimental material. Blind sampling methods were applied during the quantifications, which were carried out using computerised methods. This is now clearly explained in the Material and Methods section (see paragraph 2.9: “Quantitative analyses were carried out in a blinded fashion...”).

Reviewer 3 Report

This excellent study highlights the importance of using well characterized models of Parkinson’s disease to uncover the complex processes triggered by dopaminergic drugs with varying degrees of affinity for different dopamine receptor subtypes.   As the authors point out, patients are often treated with combinations of L-dopa and dopamine agonists with varying pharmacokinetics and overall impact on D2/D3 receptors, and it is clear that the net interactive effects of the multiple processes triggered by these combined treatments are not well understood.  This potentially important aspect of Parkinsonian drug treatment is clearly elucidated in the present study as two treatments that have similar behavioral effects in a well characterized rodent model of dyskinesia exert quite different effects on maladaptive gliovascular changes which could be consequential with respect to disease progression over time.  The dramatic effects on striatal perivascular glial reactivity of treatment with a combination of L-dopa and ropinirole, as opposed to L-dopa alone, are well demonstrated with effective immunohistochemical strategies, are novel, well discussed, potentially quite important and call for pursuit of underlying mechanisms and overall significance in future studies.   

The authors apply their excellent experience with rodent models of Parkinson’s disease in conjunction with sophisticated histochemistry, appropriate statistics, well written discussion of the results and relevant literature.   The study builds on their recent relevant study examining a range of drug doses to take examine effects of co-treatments with agonists known effects on dyskinesia.  This background allows focus on perivascular glial reactivity in the present study using well evaluated drug doses. Figures are clear and well described. Caveats are appropriately discussed.

Author Response

R: We thank the Reviewer very much for all these appreciative comments.

Reviewer 4 Report

This manuscript describes a research project that addresses the problem that patients with Parkinson's disease suffer from increased dyskinesia after several years of L-DOPA therapy. Using the 6-OHDA induced rat model of Parkinson's disease, the project investigated the extent to which such side effects can be reduced by the administration of a dopamine agonist (ropinerole) alone or by a combination therapy of L-DOPA with a dopamine agonist. For this purpose, dyskinetic movements were determined by means of scores in the test animals. It was also investigated whether the different treatment options lead to histological changes in the brains of the experimental animals. In particular, microglial population, astroglial population, microglial morphology and astroglial population as well as the tightness resp. permeability of the blood-brain barrier were investigated by immunohistochemistry, laserscanning microscopy and digital reconstruction of the substructures.

Animals treated exclusively with L-DOPA had significantly increased microglial counts in CPu. Ropinerole-treated animals alone exhibited significantly less dyskinesia. In addition, the blood-brain barrier has been shown to become more permeable after L-DOPA therapy alone.

The manuscript is well written and presents the experimental concept in a comprehensible manner.

Major concerns:

11.  Correlation analyses should still be performed correlating the measure of dyskinesis severity (AIM-score) at the end of the experiment with striatal density of albumin, microglial cell count, perivascular microglia, Iba1/RECA-1 interaction.

22. The results of the cylinder test should be presented and also correlated with the AIM score to detect a possible correlation between lesion grade and vulnerability to dyskinesia under L-DOPA therapy.

Minor:

  1. It would be nice if the methods of optical density measurement mentioned in chapter 2.4. would be described and not only referred to literature. Especially since the exact methodology of OD measurement has been described only superficially in the referenced paper as well.

  2.       The “L-DOPA” designation should be consistently followed. Currently, the manuscript contains “L-DOPA”, “L-Dopa” and “L-dopa”.

  3.       The manuscript would benefit from inset images of lesioned and sham-lesioned brains resp. brain slices containing the CPu.

  4.       The manuscript would benefit from sample images of rats exhibiting the dyskinetic behavior in question.

Author Response

R: We thank the Reviewer very much for these positive comments.

Major Concerns:

11.  Correlation analyses should still be performed correlating the measure of dyskinesis severity (AIM-score) at the end of the experiment with striatal density of albumin, microglial cell count, perivascular microglia, Iba1/RECA-1 interaction

R: We thank the reviewer for this suggestion. We have now added a supplemental Figure (Figure S3) showing linear regressions of all the above parameters on the AIM scores. Indeed, the relationship between markers of gliovascular reactivity and AIM scores is statistically significant. We have mentioned these correlations where appropriate in the Results section, referring to the corresponding plots shown in the Supplemental Figure.

  1. The results of the cylinder test should be presented and also correlated with the AIM score to detect a possible correlation between lesion grade and vulnerability to dyskinesia under L-DOPA therapy.

R: Correlations between dopaminergic lesion grade and vulnerability to L-DOPA-induced dyskinesia (LID) have been extensively investigated in papers published between 2002 and 2011 (see e.g. DOI: 10.1006/nbdi.2002.0499, DOI: 10.1016/j.nbd.2011.01.024), and extensively discussed in many review articles about LID since then (see e.g. DOI: 10.3389/fneur.2014.00242). Briefly, a threshold of DA denervation is needed for L-DOPA treatment to induce dyskinesia. However, beyond this threshold, there is no further correlation between dyskinesia severity and dopamine-denervation extent. We do not feel that it is appropriate to add such a digression to the Discussion, as it would risk to disturb the flow of the narrative. We have, however, added a sentence to paragraph 2.4 to explain that the degree of dopamine denervation did not differ between treatment groups, which we are also showing in a new supplemental Figure, see Fig. S1.

Minor:

  1. It would be nice if the methods of optical density measurement mentioned in chapter 2.4. would be described and not only referred to literature. Especially since the exact methodology of OD measurement has been described only superficially in the referenced paper as well

R: We now changed reference #13 where the O.D. methodology is well described.

  1. The “L-DOPA” designation should be consistently followed. Currently, the manuscript contains “L-DOPA”, “L-Dopa” and “L-dopa”.

R: We now uniformed the L-DOPA designation throughout the manuscript.

  1. The manuscript would benefit from inset images of lesioned and sham-lesioned brains resp. brain slices containing the CPu.

R: We have now added a supplemental Figure (Fig. S1) showing serial TH-immunostained striatal sections from the 4 different treatment groups (denervation patterns are identical between groups).

  1. The manuscript would benefit from sample images of rats exhibiting the dyskinetic behavior in question. R: We have now added a supplemental Figure (Fig. S2) showing examples of axial, limb and orolingual AIMs taken from video recordings of animals in the present study (note, however, that these dyskinetic behaviours have been extensively documented in the cited literature).

Round 2

Reviewer 2 Report

 The authors have adequately responded to all of my comments in my previous critique.

Reviewer 4 Report

The requested additions have been incorporated and suggestions for improvement have been addressed.